# Preclinical transmission of prions by blood transfusion is influenced by donor genotype and route of infection

M. Khalid F. Salamat[1⊘], A. Richard Alejo Blanco[1⊘], Sandra McCutcheon[1⊘], Kyle B. C. Tan[1], Paula Stewart[1], Helen Brown[1], Allister Smith[1], Christopher de Wolf[1], Martin H. Groschup[2], Dietmar Becher[3], Olivier Andréoletti[4], Marc Turner[5], Jean C. Manson[1], E. Fiona Houston[1]*

1 The Roslin Institute, Royal (Dick) School of Veterinary Studies, University of Edinburgh, Easter Bush, Midlothian, Edinburgh, United Kingdom, 2 Friedrich-Loeffler-Institut, Institute of Novel and Emerging Infectious Diseases, Greifswald, Germany, 3 Micromun GmbH, Greifswald, Germany, 4 UMR INRA ENVT 1225, Interactions Hôtes Agents Pathogènes, Ecole Nationale Vétérinaire de Toulouse, Toulouse, France, 5 Scottish National Blood Transfusion Service (SNBTS), The Jack Copland Centre, Edinburgh, United Kingdom

⊘ These authors contributed equally to this work.
* fiona.houston@roslin.ed.ac.uk

**Data Availability Statement:** All relevant data are within the manuscript and its Supporting Information files.

## Abstract

Variant Creutzfeldt-Jakob disease (vCJD) is a human prion disease resulting from zoonotic transmission of bovine spongiform encephalopathy (BSE). Documented cases of vCJD transmission by blood transfusion necessitate on-going risk reduction measures to protect blood supplies, such as leucodepletion (removal of white blood cells, WBCs). This study set out to determine the risks of prion transmission by transfusion of labile blood components (red blood cells, platelets, plasma) commonly used in human medicine, and the effectiveness of leucodepletion in preventing infection, using BSE-infected sheep as a model. All components were capable of transmitting prion disease when donors were in the preclinical phase of infection, with the highest rates of infection in recipients of whole blood and buffy coat, and the lowest in recipients of plasma. Leucodepletion of components (<$10^6$ WBCs/ unit) resulted in significantly lower transmission rates, but did not completely prevent transmission by any component. Donor *PRNP* genotype at codon 141, which is associated with variation in incubation period, also had a significant effect on transfusion transmission rates. A sensitive protein misfolding cyclic amplification (PMCA) assay, applied to longitudinal series of blood samples, identified infected sheep from 4 months post infection. However, in donor sheep (orally infected), the onset of detection of PrP$^{Sc}$ in blood was much more variable, and generally later, compared to recipients (intravenous infection). This shows that the route and method of infection may profoundly affect the period during which an individual is infectious, and the test sensitivity required for reliable preclinical diagnosis, both of which have important implications for disease control. Our results emphasize that blood transfusion can be a highly efficient route of transmission for prion diseases. Given current uncertainties over the prevalence of asymptomatic vCJD carriers, this argues for the

**Funding:** This work represents independent research commissioned and funded by the Policy Research Programme of the Department of Health and Social Care (https://www.nihr.ac.uk/explore-nihr/funding-programmes/policy-research.htm). The award (reference 007/0162) was made to EFH, JCM & MT. The funders had no role in study design, data collection and analysis, decision to publish, or preparation of the manuscript.

**Competing interests:** The authors have declared that no competing interests exist.

maintenance and improvement of current measures to reduce the risk of transmission by blood products.

## Author summary

Variant Creutzfeldt-Jakob disease (vCJD) resulted from zoonotic transmission of bovine spongiform encephalopathy (BSE), and has also been transmitted by blood transfusion. One of the most important risk reduction measures introduced by human transfusion services to safeguard the blood supply is leucodepletion (removal of white blood cells) of blood components. This study represents the largest experimental analysis to date of the risks of prion infection associated with transfusion of labile blood components, and the effectiveness of leucodepletion in preventing transmission. Using a BSE-infected sheep model, we found that red blood cells, platelets and plasma from preclinical donors were all infectious, even after leucodepletion, although leucodepletion significantly reduced transmission rates. In addition, the time course of detection of prions in blood varied significantly depending on the route and method of infection. This has important implications for the risk of onward transmission, and suggests that further improvements in sensitivity of diagnostic tests will be required for reliable preclinical diagnosis of vCJD and other prion diseases. The results of this study support the continuation of current measures to reduce the risk of vCJD transmission by blood products, and suggest areas for further improvement.

## Introduction

The emergence of a novel human prion disease, variant Creutzfeldt-Jakob disease (vCJD), in the UK during the 1990s posed significant challenges for public health policy [1]. Although it was soon recognized that vCJD was the result of human exposure to meat products from cattle infected with bovine spongiform encephalopathy (BSE) [2,3], the probable size of the epidemic and the risks of ongoing human-to-human transmission of infection were initially very uncertain. Since there are on average 1.5–2 million units of blood donated annually in the UK, the possibility of vCJD transmission *via* blood transfusion was a major concern.

According to the prion hypothesis, diseases such as vCJD and BSE are caused by "proteinaceous infectious particles" (= prions), which consist of a misfolded form (designated $PrP^{Sc}$) of a normal host protein, $PrP^{C}$. The aetiology of the most common form of human prion disease, sporadic CJD (sCJD), is uncertain, but iatrogenic transmission of sCJD by, e.g. injection of cadaveric growth hormome extracts, dura mater grafts, has been demonstrated. However, epidemiological studies have so far failed to find any definitive evidence of sCJD transmission by blood transfusion [4]. When vCJD was initially characterised, there were strong indications that it might behave differently to sCJD with respect to transmission by transfusion. Since infection appeared to have been acquired from ingestion of contaminated food, it was possible that prions could enter the bloodstream during their dissemination from the gut to central nervous system. In addition, unlike sCJD patients, vCJD cases had readily detectable deposits of $PrP^{Sc}$ in secondary lymphoid tissues (e.g. tonsil, spleen, lymph nodes) [5,6], suggesting a greater risk of infection entering the blood on recirculating lymphocytes. Studies in rodent models of prion infection also demonstrated low levels of infectivity in blood, the majority of which appeared to be associated with leucocytes (white blood cells) [7–9]. Therefore, one of

the first UK blood safety measures in response to vCJD was the introduction in 1998–1999 of universal leucodepletion of blood components, although definitive evidence of its efficacy was lacking at that time.

From 2000 onward, studies in sheep clearly demonstrated that prion infection could be efficiently transmitted following transfusion of blood from prion-infected donors, well before they showed clinical signs of disease [10–12]. The Transfusion Medicine Epidemiology Review (TMER), which was set up to determine whether there was any link between vCJD and blood transfusion, subsequently identified 3 cases of vCJD in recipients of blood products from donors who went on to develop vCJD [13,14]. In addition, one transfusion recipient who was followed up showed post mortem evidence of PrP^Sc deposition in the spleen, although the cause of death was unrelated to vCJD [15]. These cases confirmed that vCJD could be transmitted by blood transfusion, and prompted a re-evaluation of vCJD infection risks associated with all blood products.

Although the incidence of vCJD is now very low, and there have been no new transfusion-related cases since 2003, concerns about the safety of blood products remain. These arise from surveys of samples removed during surgical appendectomy, which showed that an estimated 1 in 4000 to 1 in 2000 of the UK population might have abnormal PrP deposition in the lymphoid follicles of the appendix, similar to that seen in vCJD patients [16,17]. It is possible that these individuals are subclinically infected (i.e. they will never develop symptoms of vCJD within their natural lifespan), and their potential to transmit infection by blood transfusion or other iatrogenic routes is currently unknown. In addition, the last confirmed vCJD case in the UK occurred in an individual heterozygous at *PRNP* codon 129 (129MV) [18]; all previous cases that were tested having been 129MM. Since the *PRNP* 129MV genotype has been associated with prolonged incubation periods in other acquired human prion diseases (e.g. kuru, iatrogenic CJD), this heightened concerns of a possible second wave of vCJD cases.

This study was set up to determine the distribution of prion infectivity in labile blood components commonly used in transfusion medicine, and the effectiveness of leucodepletion of blood components in preventing transmission of infection, using sheep experimentally infected with BSE as a model. The sheep model offered several advantages: sheep could be infected with BSE by the oral route, they showed PrP^Sc deposition in lymphoid tissues similar to that seen in vCJD patients, and their similarity to humans in body size allowed collection of equivalent volumes of blood that could be processed to yield comparable components. Interim findings indicated that red cells, platelets and plasma prepared from preclinical donors were all capable of transmitting infection, and that leucodepletion reduced, but did not eliminate, the risk of transmission [19]. In this paper, we present the full results of the study after follow-up of remaining transfusion recipients for up to 11 years (close to their natural lifespan). In addition, we show that infectivity titres in blood components reflect transfusion transmission efficiency, report the outcomes of secondary transfusions (with primary recipients as donors), and show how PrP^Sc detection in blood by protein misfolding cyclic amplification assay (PMCA) varies over the time course of infection.

## Materials and methods

### Ethics statement

The sheep experiments were reviewed and approved by animal welfare and ethical review committees at the Institute for Animal Health (Compton) and The Roslin Institute, and carried out under the authority of Home Office Project Licences (references: 30/2282, 60/4143, 70/8595).

### Experimental infection of sheep

Cheviot sheep were sourced from the Defra scrapie-free flock, which was derived from sheep imported from New Zealand. All sheep had the *PRNP* genotype ARQ/ARQ (using standard terminology for amino acids encoded by codons 136, 154 and 171 of the *PRNP* gene), which is most susceptible to BSE. They did not have additional polymorphisms at codons 112 and 168 (M112T and P168L, respectively) that have been associated with reduced susceptibility to prion disease in sheep [20,21]. However, they did have relatively high frequencies of the L141F polymorphism at codon 141, which at the time was not known to influence susceptibility to BSE.

The experimental design has previously been described in detail (summarized in Fig 1) [19]. Briefly, donor sheep (n = 40) aged between 6–12 months were experimentally infected with BSE by administration of an oral dose of 5 g of BSE-infected cattle brain homogenate. Negative control sheep (n = 10) were orally dosed with 5 g of uninfected cattle brain homogenate. One BSE-challenged sheep and one negative control sheep were culled due to intercurrent disease before blood could be collected for transfusion, and were therefore excluded from the study.

Recipient sheep (n = 244) aged between 6–24 months were transfused with blood components prepared from infected and negative control donors as previously described [19,22]. Eighteen recipients (transfused with either whole blood or buffy coat from orally infected donors) were used as donors of whole blood (1 unit = 450 ml ± 10% v/v) for transfusion into secondary recipients.

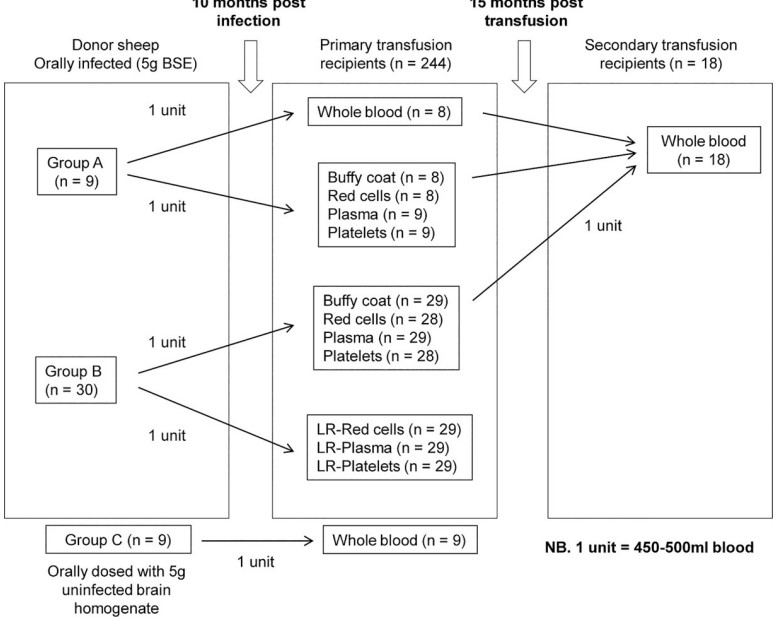

**Fig 1. Schematic representation of the experimental design of the sheep blood transfusion experiments.** Sheep used as blood donors were orally infected with 5g BSE-infected cattle brain homogenate. Two units of blood (450-500ml each) were collected from each donor at 10 months post-infection (mpi). Group A–one unit was transfused as whole blood; the second unit was processed to components, before transfusion to individual recipient sheep (total of 5 recipients per donor). Group B–both units were processed to components, and one set of components was leucodepleted, before transfusion to recipients (total of 7 recipients per donor). Group C–negative control donors were orally dosed with 5g normal cattle brain homogenate (i.e. mock-infected), and one unit of whole blood was collected from each sheep at 10 mpi and transfused into recipients. To assess secondary transmission of infection by transfusion, one unit of whole blood was collected from 18 primary recipients of whole blood (n = 8) and buffy coat (n = 10) at 15 mpi, and transfused into individual secondary recipients.

Procedures for housing, management and clinical scoring of sheep are as previously described [22]. Sheep with positive TSE clinical scores were euthanized once they reached defined humane end points. Other sheep were euthanized because of intercurrent disease or welfare concerns, or at the end of different phases of the project. For all sheep, the survival period (SP) was calculated as the interval between the date of infection and the date of euthanasia.

## Preparation of blood components for transfusion

The preparation and validation of blood components has previously been described in detail [19]. Briefly, two units of whole blood (WB; 1 unit = 450 ml ± 10% v/v) were collected from each donor sheep into blood packs (Fresenius Hemocare NPBI), containing CPD.A1 anticoagulant. Components were separated by centrifugation and extraction using a Compomat G4 (Fresenius Hemocare) optimised for the separation of sheep blood components, to yield platelet-rich plasma, red cells and buffy coat. Platelet-rich plasma was further centrifuged to produce platelet-poor plasma and platelet concentrate. Leucodepletion of components was performed using in-line leucoreduction filters (Fresenius/NPBI).

Prior to transfusion, 10–20 ml of each component was sampled for analysis and archive storage/bioassay. For each component, the volume was estimated based on weight (adjusted for specific gravity). Numbers of leucocytes in non-leucodepleted and leucodepleted components were measured by flow cytometry, using a Leucocount kit (BD Biosciences). Numbers of platelets in platelet concentrates and plasma fractions were counted manually using a haemocytometer. In whole blood, red cells and buffy coat, the haematocrit was measured, using a haematocrit centrifuge and reader (Hawksley), to determine the distribution of plasma in different components.

## Experimental inoculation of TgShpXI mice with sheep blood components

Sheep blood samples were stored at -80 °C, and shipped on dry ice to Micromun, Greifswald, Germany. Following microbial screening and heat treatment (70 °C for 15 min), samples were inoculated into 6 week old TgShpXI mice [23]. Each mouse was inoculated intracerebrally with 25 µl sample (undiluted or diluted 1:1 with sterile PBS) under general anaesthesia. Inoculated mice were housed in groups of up to ten in a Containment Level 3 (CL3) facility and monitored for development of clinical signs up to 700 days post-infection. Brains were collected at necropsy from all mice, including intercurrent deaths, and tested for the presence of PrP$^{Sc}$ using an ELISA kit (BetaPrion BSE EIA Test Kit; AJ Roboscreen Gmbh, Leipzig, Germany), according to the manufacturer's instructions.

## Collection of blood samples for archiving

Blood samples were collected from most donor sheep and selected primary and secondary transfusion recipients before infection and at regular intervals post-infection until they developed clinical signs or were culled for other reasons. Approximately 100 ml of blood was collected into a small collection bag containing 1.5 ml 0.5 M EDTA, pH 8.0 as anticoagulant. Following removal of 16ml whole blood, the remaining blood was separated into plasma, buffy coat and red cell fractions by centrifugation at 1000 x g for 30 minutes at ambient temperature. Aliquots of whole blood, plasma, red cells and buffy coat were transferred to cryovials and stored at -80 °C.

## Sample collection at necropsy

Tissue samples were collected from all sheep, including intercurrent deaths, immediately following euthanasia. The brain was divided in half longitudinally, and half fixed in neutral

buffered formalin (NBF), whilst the other half was frozen at -80 °C. Samples of palatine tonsil, spleen, ileal Peyer's patch, mesenteric lymph node and prescapular lymph node were also fixed in NBF, and frozen at -80 °C, prior to further analysis. All frozen samples were collected using separate disposable instruments and using aseptic precautions.

## Immunohistochemistry for PrP^Sc detection

Fixed tissue samples were immersed in 98% (v/v) formic acid for 1 hour to reduce infectious titres of BSE, and processed according to standard histopathological techniques, as previously described [24]. Immunohistochemistry was performed on tissue sections by well-established methods, as previously described [25,26]. The primary antibodies used were mouse monoclonal antibodies BG4 (raised against recombinant bovine PrP; epitope amino acids [47]PGGNRYP PQGG[57] and [89]GGGGWGQGGSH[99]) at 1.25 μg/ml, and ROS-IH9 (raised against recombinant sheep PrP residues 94–233; epitope amino acids [140]PLIHFG[145]) at 0.1 μg/ml [25]; the chromogen was Vector NovaRed (Vector Laboratories).

## Western blotting for PrP^Sc detection

Samples of frozen tissues were thawed, weighed and homogenized in phosphate buffered saline, pH 7.4 (PBS) to give 10% (brain) or 20% (lymphoid tissues) w/v homogenates. Western blotting for PrP^Sc detection was performed following proteinase K digestion and sodium phosphotungstic acid (NaPTA) precipitation of tissue homogenates, as previously described [27], using the primary anti-PrP monoclonal antibodies ROS-BC6 (raised against recombinant sheep PrP residues 94–233; epitope amino acids [144]FGNDYEDRYYR[154]) and ROS-IH9, both at 0.5 μg/ml [25].

## Protein misfolding cyclic amplification (PMCA) for detection of PrP^Sc

We used a microplate-based, miniaturized bead serial PMCA protocol, as published previously [28]. Brains from TgShpXI transgenic mice that over-express sheep PrP^C from the ARQ *PRNP* allele (with leucine at codon 141) [23] were used as the source of PrP^C (termed "substrate"). The mouse brains were homogenised in PMCA buffer (PBS pH 7.2, 0.25% v/v Triton X100, 150 mM NaCl, cOmplete Protease Inhibitor (Roche) at 4 °C) to provide a 10% (w/v) homogenate, using a pestle and mortar (Dounce) glass homogenizer. Samples of tissues from BSE-infected sheep and controls were homogenized in PMCA buffer using an Omni electronic homogeniser with disposable probes (Camlab), to yield 10% w/v homogenates (termed "seed"). The substrate and seed homogenates were passed through 25G needles, and stored in aliquots at −80°C. Sheep blood samples were not homogenized, but were thawed from -80 °C, and used either undiluted, or diluted in PMCA buffer, as seed preparations.

PMCA reactions were performed in Axygen 96 well PCR microtitre plates, with 45 μl of substrate homogenate and one Teflon bead (2.381 mm, Marteau & Lemarie, France) per well, seeded with 5 μl seed homogenate or blood sample; each sample was run in duplicate. In some experiments, 5% (w/v) dextran sulphate (Sigma-Aldrich) solution was added to each reaction to give a final concentration of 0.5%. Positive controls run in each experiment consisted of 10-fold serial dilutions of pooled brain homogenate from five BSE positive sheep, which had previously been titrated by intracerebral inoculation to end-point dilution in gene-targeted transgenic mice expressing sheep ARQ PrP. Negative controls consisted of reactions seeded with tissue homogenates or blood samples from negative control (mock-infected) sheep, substrate only, and/or substrate with added dextran sulphate.

The microplate was sealed using Axygen 8-strip domed PCR tube caps and Parafilm, and placed in a microplate horn attached to a programmable Misonix Q700 sonicator (QSonica).

The temperature in the microplate horn was maintained at 37 °C using a water recirculation system incorporating a water bath and peristaltic pump. A total of 96 PMCA cycles (constituting one round of amplification) were performed, with each cycle comprising 10 sec of sonication (at an amplitude of 50–65%) and 14 min 50 sec incubation. After each round of amplification, 5 µl reaction product from each well was mixed with 45 µl fresh substrate in a new microtitre plate, and PMCA amplification repeated.

Aliquots of reaction products from each PMCA round were digested with 0.1 mg/ml proteinase-K (Qiagen) for 1h at 37˚C, and detected by Western blotting, as previously described [27–29]. Briefly, electrophoresis and blotting of the equivalent of 3.6 µL of each PK-digested PMCA product was performed using standard protocols. Blots were probed with the monoclonal antibody ROS-BC6 (0.25 µg/ml), developed using Clarity Western ECL Substrate (Bio-Rad) and visualised on Amersham Hyperfilm ECL. Molecular weight markers used were Precision Plus Protein Prestained Standards (Bio-Rad).

### Statistical analysis

Transmission rates were compared between blood components using a random effects Poisson model. This fitted blood component, donor %SP (stage of survival period = time from infection to blood collection x 100/time from infection to culling due to clinical signs) and recipient genotype as fixed effects, and donor as a random effect. Fitting donor as random allows for potential correlations in results from the same donor, and fitting %SP and genotype adjusts results for differences in these effects between the blood components and improves the model efficiency. The data were too sparse to assess whole blood within this model, and comparisons between whole blood and other components were made using Fisher's exact tests.

The effect of leucodepletion was assessed using a random effects Poisson model with leucodepletion (yes/no), component (plasma, platelets, red cells), leucodepletion by component interaction, and %SP as fixed effects; and donor as a random effect. This allowed both the overall effect of leucodepletion, and its effect for each blood component, to be assessed.

Donor survival periods were compared between donor genotypes using t-tests, and transmission rates were compared between donor genotypes using a Fisher's exact test.

## Results

### Distribution of prion infectivity in blood

The majority of donor sheep (31/39) orally dosed with BSE-infected brain homogenate developed typical clinical signs (e.g. behaviour changes, pruritus, head tremors, ataxia), and two were culled as a result of intercurrent disease. All thirty-three animals were confirmed as being infected with BSE by detection of PrP$^{Sc}$ in brain and lymphoid tissues tested post mortem by Western blotting and/or immunohistochemistry (IHC). During the course of our experiment, a previously unrecognised association was identified between the survival period of infected sheep and a polymorphism at *PRNP* codon 141 (resulting in a leucine to phenylalanine amino acid substitution) [30]. In animals that showed clinical signs and tested positive for PrP$^{Sc}$ in IHC/Western blot, mean survival periods (± standard deviation) were 625 ± 111 days for 141LL (n = 11), 861 ± 142 days for 141FF (n = 7) and 1253 ± 190 days for 141LF sheep (n = 13), respectively, and these differences were statistically significant when analysed using t-tests (p<0.001).

Five of the remaining six donors were culled after showing signs of ataxia, which is one of the clinical signs observed in BSE-infected sheep. However, analysis of their brain and lymphoid tissues for PrP$^{Sc}$ by IHC and Western blotting gave negative results. Therefore, they were considered to be uninfected, which is consistent with earlier studies in which a

**Table 1. Efficiency of transmission of BSE by different blood components following transfusion in sheep.**

| Component | No. sheep transfused | No. sheep negative (%) | No. sheep BSE positive (%) |
|---|---|---|---|
| Whole blood | 8 | 4 (50%) | 4 (50%) |
| Buffy coat | 31 | 18 (57%) | 13 (42%) |
| Platelets | 31 | 21 (68%) | 10 (32%) |
| Red cells | 29 | 21 (73%) | 8 (27%) |
| Plasma | 32 | 26 (81%) | 6 (19%) |
| Platelets (leucodepleted) | 22 | 21 (95%) | 1 (5%) |
| Red cells (leucodepleted) | 22 | 19 (86%) | 3 (14%) |
| Plasma (leucodepleted) | 23 | 21 (91%) | 2 (9%) |

proportion of ARQ/ARQ sheep ≥6 months old remained uninfected following oral challenge with BSE [26,31]. Neuropathological examination of three affected sheep found evidence of axonal degeneration affecting the spinal cord, consistent with a toxic or metabolic cause, but further investigation failed to establish a definitive diagnosis. Subsequent similar cases in small numbers of recipients were therefore classified as "idiopathic ataxia". Details of the clinical scores, cause of death and results of IHC and Western blotting on tissues from all donor sheep are presented in S1 Table.

Table 1 shows the numbers of recipient sheep that showed evidence of BSE infection (clinical signs and/or positive results from Western blotting and IHC on brain and lymphoid tissues) following transfusion of each blood component. For all animals identified as BSE positive, even in the absence of clinical signs, PrP$^{Sc}$ was detected in the brain as well as lymphoid tissues, indicating that they were at least in the late pre-clinical phase of infection. Similar to the donor sheep, the survival period of recipients that showed clinical signs and tested positive for PrP$^{Sc}$ in IHC/Western blot was associated with *PRNP* codon 141 genotype. The mean survival periods (± standard deviation) were 727 ± 122 days for 141FF sheep (n = 15) and 1021 ± 170 days for 141LF sheep (n = 22), and this difference is statistically significant when analysed by t-test (p <0.001). The two 141LL recipients that developed clinical signs and were confirmed as BSE-positive by IHC/Western blot had survival periods of 513 and 595 days, respectively.

In a small number of recipients, atypical scrapie was identified during post mortem examination of tissues. These cases could be clearly distinguished from BSE infection based on the distribution of neuropathological lesions/PrP$^{Sc}$ (predominantly in cerebral cortex and cerebellum versus brainstem for BSE), lack of detectable PrP$^{Sc}$ in lymphoid tissues, and a distinctive low molecular weight (<15 kDa) band on Western blot of PK-digested PrP$^{Sc}$. They were therefore classified as BSE negative. Details of the clinical scores, cause of death and results of IHC and Western blotting on tissues from all recipient sheep are presented in S2 Table.

The proportion of BSE-infected recipients for each blood component (transmission rate) gives an indication of which components are associated with the highest risk of transmission, reflecting the probable levels or titres of infectivity (Table 1). The figures presented do not include recipients of blood components from the six donors considered to be uninfected following oral dosing with BSE (see above). We also excluded recipients which died or were culled for welfare reasons <12 months following transfusion, since we cannot rule out the possibility that they may have been infected, even if tests on their tissues were negative.

The results show that all blood components are capable of transmitting BSE infection, but with varying efficiency. The highest transmission rates were found in recipients of whole blood (50%, i.e. half of recipients of whole blood became infected) and buffy coat (42%), while the lowest transmission rate (19%) was in recipients of plasma, with platelets and red cells

giving intermediate values of 32% and 27%, respectively. Statistical analysis shows that transmission rates were significantly higher for buffy coat than for plasma (p<0.001), platelets (p = 0.012) and red cells (p = 0.005); and significantly higher for whole blood than for plasma (p = 0.03). However, the power of tests involving whole blood was very low as there were only 8 recipients. Recipient *PRNP* codon 141 genotype was not associated with a statistically significant effect on transmission of infection.

In order to determine whether the variation in transmission efficiency reflects variation in infectivity titres in different blood components, samples of the same components were inoculated intracerebrally into groups of transgenic mice over-expressing sheep ARQ PrP (TgshpXI). A pilot study in which serial tenfold dilutions of a BSE-infected sheep brain homogenate were injected into groups of TgShpXI mice established that the limit of detection of infectivity in this mouse line was approximately $2.5 \times 10^{-3}$ $ID_{50}$, equivalent to a $10^{-7}$ dilution of brain homogenate (S3 Table).

Only nine blood components (5 buffy coat, 2 whole blood, 2 platelets), derived from five individual donor sheep, gave positive transmissions in the mouse bioassay (Table 2). Infectious titres for these components were calculated by limiting dilution titration (where distribution of infectivity into individual inoculations is assumed to follow a Poisson distribution), as previously described [32]. The titres of infectivity varied considerably between donor sheep, but were consistently highest in buffy coat (mean ± SD = 13.1 ± 9.9 ID/ml; range from 2.8 to 29.2 ID/ml), with comparatively lower titres in whole blood (1.4 and 1.5 ID/ml) and platelets (1.5 and 3.2 ID/ml).

Each of the components that transmitted infection in mice also transmitted infection to the respective recipient sheep at 10 months post-infection. Indeed, all blood components, including leucodepleted components, from these five donor sheep transmitted infection to their respective recipients, suggesting that they had some of the highest titres of blood infectivity among donors. Thus, the TgShpXI mouse bioassay was much less sensitive in detecting infectivity in blood components than sheep blood transfusion, probably because of the difference in volumes that could be injected. Titres of infectivity in plasma, red cells and leucodepleted components were below the limits of detection of the bioassay. Nevertheless, the results confirm and extend the outcomes of the sheep transfusion experiments. Positive transmissions in mice resulted from inoculation of buffy coat, whole blood and platelets, which were also the components that produced the highest transmission rates in sheep transfusions. The highest titres of infectivity were found in buffy coat, suggesting that the majority of blood-borne infectivity is associated with white blood cells.

**Table 2. Titres of infectivity in sheep blood components estimated by bioassay in TgshpXI mice.**

| Donor sheep ID | Blood component | Number of positive mice/ number inoculated | Titre (ID/ml) |
|---|---|---|---|
| N257 | Buffy coat | 2/30 | 2.8 |
| | Whole blood | 1/30 | 1.4 |
| N236 | Buffy coat | 7/25 | 13.1 |
| N233 | Buffy coat | 14/27 | 29.2 |
| | Whole blood | 1/27 | 1.5 |
| | Platelets | 1/27 | 1.5 |
| N251 | Buffy coat | 5/26 | 8.5 |
| N261 | Buffy coat | 7/27 | 12.0 |
| | Platelets | 2/26 | 3.2 |

Each mouse was inoculated intracerebrally with 25μl of the indicated component (undiluted) collected in citrate-phosphate-dextrose with adenine (CPD.A1) as anticoagulant.

## Effect of leucodepletion on transmission of prions by transfusion

For the majority of BSE-infected donors (29 sheep), paired samples of leucodepleted and non-leucodepleted red cell concentrate, platelets and plasma were prepared, and each component was transfused into an individual recipient (Fig 1, Group B). Taking the results for all transfusions carried out in Groups A and B together, transmission rates were 14%, 5% and 9% in recipients of leucodepleted red cells, platelets and plasma, respectively, compared to 30%, 32% and 19% in recipients of the equivalent non-leucodepleted components (Table 1). Combining the data for all three components, the lower transmission rate for leucoreduced components was statistically significant (p = 0.01). Thus, leucodepletion of blood components substantially reduces, but does not eliminate, the risk of transmission of BSE by blood transfusion in sheep.

When the analysis was restricted to smaller numbers of recipients transfused with matched leucodepleted and non-leucodepleted components from the same BSE-infected donor, a similar statistically significant difference in transmission rate for leucodepleted components was observed when data for red cells, platelets and plasma were combined (p = 0.02; Table 3). There was insufficient power to carry out statistical tests to compare individual leucodepleted and non-leucodepleted components.

## Transmission to secondary transfusion recipients

To test whether sheep infected with BSE *via* blood transfusion were in turn able to transmit the infection by transfusion, one unit of whole blood (450ml ± 10%) was collected from 18 primary recipients at 15 months post-infection and transfused into individual secondary recipients. Details of the outcomes of the experiment for individual pairs of primary (donor) and secondary recipients are shown in S4 Table. At the time of blood collection, it was not possible to determine which of the primary recipients acting as donors in this experiment were infected with BSE. Eight out of the eighteen primary recipients used as donors in this experiment subsequently developed clinical disease and were confirmed as being BSE-positive by post mortem testing, with survival periods ranging from 16 to 40 months post-infection (mean ± SD = 24 ± 8 months). Details of the clinical scores and results of IHC and Western blotting on tissues from all secondary recipient sheep are included in S5 Table. Blood collection occurred between 1 month and 25 months before sheep were culled following the onset of clinical signs. Secondary recipients of whole blood transfusions from these eight infected primary recipients also developed clinical disease and were confirmed BSE-positive, with survival periods ranging from 21 to 39 months post-infection (mean ± SD = 31 ± 5 months). Therefore, the transmission rate from infected donors was 100% in this experiment. The remaining ten primary recipients and their corresponding secondary recipients did not show clinical signs typical of BSE during periods of observation of between 33 and 132 months post transfusion, although a number of them were scored positive for idiopathic ataxia, and one of the primary recipients was diagnosed as having atypical scrapie. However, none of the post mortem tests of brain and lymphoid tissues from these animals revealed evidence of BSE infection. This result demonstrates that individuals infected with prion disease by blood transfusion are themselves highly likely to transmit infection if they donate blood.

**Table 3. Transmission of BSE by transfusion of leucodepleted and non-leucodepleted blood components from the same donor.**

| Component | Non-leucodepleted | | Leucodepleted | |
|---|---|---|---|---|
| | No. BSE positive recipients (%) | No. BSE negative recipients (%) | No. BSE positive recipients (%) | No. BSE negative recipients (%) |
| Plasma | 3 (14%) | 19 (86%) | 2 (9%) | 20 (91%) |
| Platelets | 6 (30%) | 14 (70%) | 1 (5%) | 19 (95%) |
| Red cells | 5 (25%) | 15 (75%) | 3 (15%) | 17 (85%) |

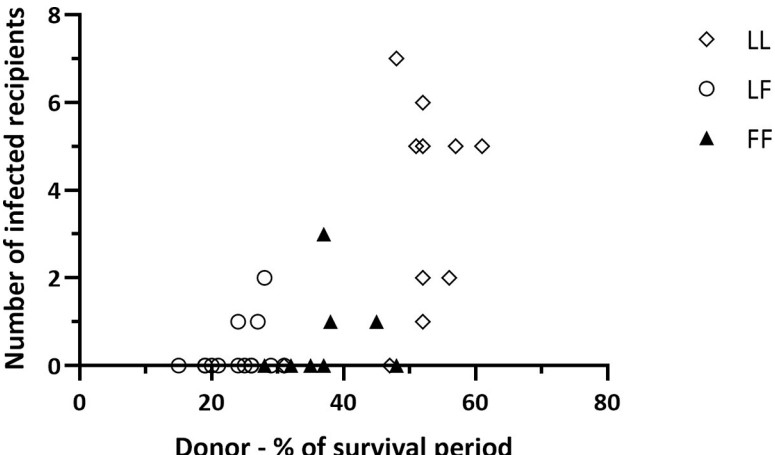

**Fig 2. Relationship between donor genotype, stage of infection and transfusion transmission rate.** Each point on the chart represents an individual infected donor sheep, showing the number of corresponding transfusion recipients infected (all components) plotted against the stage of survival period at which blood was collected (%SP = time from infection to blood collection x 100/time from infection to culling). The donor *PRNP* codon 141 genotypes are represented by different symbols: 141LL–open diamonds; 141FF–closed triangles; 141LF–open circles. The longer survival periods of 141FF and 141LF compared to 141LL sheep mean that they were at an earlier stage of infection when they donated blood for transfusion (10 months post infection for all donors).

## Probability of transfusion transmission of prions is influenced by donor *PRNP* genotype

Although the majority of donor sheep were infected with BSE, not all of them transmitted the infection to primary transfusion recipients. Blood was collected at 10 months post infection, which was estimated to be approximately halfway through the survival period based on previous experiments in which *PRNP* genotype ARQ/ARQ sheep were orally infected with similar doses of BSE-infected cattle brain homogenate (typically 20–24 months). However, due to the variation in survival periods associated with *PRNP* codon 141 genotype, 10 months post-infection actually represented between 15% and 61% of the total survival period for individual sheep (Fig 2).

Each donor sheep contributed either five or seven different blood components, depending on the experiment (Fig 1), that were each transfused into individual recipients. Fifteen infected donors out of thirty-three (45%) transmitted infection to one or more of the corresponding recipients ("transmitters"), while eighteen (55%) did not transmit infection to any recipients ("non-transmitters"). The probability of transmission correlated with the donor *PRNP* codon 141 genotype. As shown in Table 4, out of fifteen transmitters, nine (60%) were 141LL, three (20%) were 141FF and three (20%) were 141LF, whereas out of eighteen non-transmitters, two (11%) were 141LL, five (27%) were 141FF and eleven (61%) were 141LF. The differences in transmission rate between the donor genotypes were statistically significant (p = 0.01), with the highest rate of transmission for 141LL donors.

**Table 4. Association of donor *PRNP* genotype with transfusion transmission.**

| Donor codon 141 *PRNP* genotype | Transmitters | Non-Transmitters |
|---|---|---|
| | Number (%) | Number (%) |
| LL | 9 (60%) | 2 (11%) |
| FF | 3 (20%) | 5 (28%) |
| LF | 3 (20%) | 11 (61%) |

**Table 5. Detection of PrP$^{Sc}$ by PMCA in buffy coat samples BSE-infected donor sheep at 10 months post infection (preclinical).**

| Sheep donor group | Number tested | PMCA results (percentage of positive PMCA results in repeat experiments[a]) | | | |
|---|---|---|---|---|---|
| | | Consistently positive (75–100%) | Intermittently positive (20–40%) | Occasionally positive (13–14%) | Negative |
| "Transmitters" | 12 | 9[b] | 3 | 0 | 0 |
| "Non-transmitters" | 9 | 0 | 0 | 2 | 7 |
| Exposed, uninfected | 2 | 0 | 0 | 0 | 2 |
| Negative control | 3 | 0 | 0 | 0 | 3 |

a Each sample was tested in 3–10 repeat experiments, and the number of positive results expressed as a percentage of the number of experimental repeats.

b Includes the five sheep (N257, N236, N233, N251, N261) in which buffy coat samples collected at 10 months post infection transmitted BSE to TgShpXI mice (Table 2)

The most likely explanation for these results is that the probability of transmission is related to the stage of the survival period (i.e. time from infection to blood collection x 100/time from infection to culling) reached by donor sheep at the time of blood collection (Fig 2). Since 141LL sheep have the shortest survival periods, they were on average about halfway through their survival period (mean 52%; range 47% - 61% of survival period) at this point. In contrast, 141LF sheep, which have the longest survival periods, were on average less than a third of the way through their survival period (mean 25%; 15% - 30% of survival period) at the same time point, and 141FF donors were at an intermediate stage (mean 35%; 28% - 48% of survival period). Therefore, at 10 months post infection the majority of 141FF and 141LF sheep, being at an earlier stage of infection than 141LL sheep, may either have no prions in blood, or blood infectivity titres that are too low to transmit infection. It is also possible that titres of blood infectivity of 141FF and 141LF sheep never reach similar levels to those of 141LL animals.

## Time course of detection of PrP$^{Sc}$ in blood of prion-infected sheep

To further examine how titres of infectivity in blood change over the time course of infection, we used the archive of samples collected from donor and recipient sheep at intervals throughout the preclinical and clinical stages of infection. Since the low sensitivity of the mouse bioassay did not allow assessment of infectivity in the majority of preclinical blood samples (S6 Table), we used a sensitive *in vitro* method for amplification of PrP$^{Sc}$ (protein misfolding cyclic amplification, or PMCA) as an alternative approach to prion detection in blood of BSE-infected sheep [28,29].

Initial optimization experiments demonstrated the sensitivity of the PMCA assay, with the limit of detection being reached at dilutions of $10^{-8}$ to $10^{-10}$ (equivalent to approximately 0.5–50 pg brain) of the reference BSE-infected sheep brain pool homogenate, and showed that sheep blood components did not have a significant inhibitory effect on the assay (S1 Fig). Further optimization using whole blood, buffy coat and plasma samples from sheep clinically affected by BSE demonstrated that the sensitivity of detection of PrP$^{Sc}$ was highest when buffy coat samples were used to seed PMCA reactions (S2 Fig and S2 and S7 Tables).

To determine whether there was a correlation between detection of PrP$^{Sc}$/seeding activity by PMCA and BSE infectivity in blood, we tested buffy coat samples from donor sheep at the same time point that blood was collected for transfusion (10 months post-infection). As explained above, some donor sheep transmitted infection to one or more of their respective transfusion recipients (thus were termed "transmitters") and others did not transmit infection to any recipients ("non-transmitters"). The majority of samples (9/12) from "transmitters" gave positive results in 75–100% of repeat PMCA experiments, while the remaining three gave positive PMCA results in 20–40% of repeat experiments (Table 5). Two out of nine samples from "non-transmitters" gave positive PMCA reactions in one out of six and one out of seven

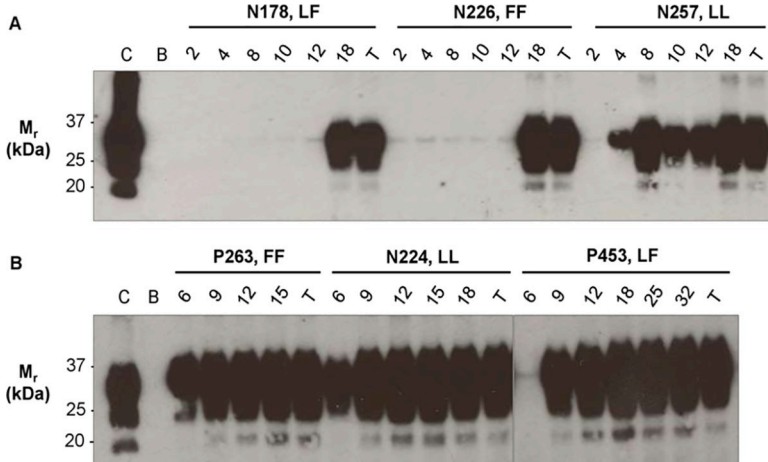

**Fig 3. Detection of PrP^Sc in preclinical buffy coat samples from BSE-infected donor and recipient sheep.**
Longitudinal series of buffy coat samples from three donors (N178, N226, N257; upper panel A) and three recipients (P263, N224, P453; lower panel B) were used undiluted to seed PMCA reactions. All three possible *PRNP* codon 141 genotypes (LL, FF, LF) are represented. Numbers at the top of each panel indicate the stage of infection at each sampling point in months post-infection (mpi), with T (terminal) representing the final sampling point when sheep were culled with clinical signs of disease. The image shows representative results from a single experimental run after 2 rounds (R2) of PMCA (dextran sulphate added to final concentration of 0.5% w/v). C–positive control; BSE-infected sheep brain homogenate (PK digested; 1.7mg brain equivalent). B–blank lane.

repeat experiments respectively, whilst the remaining 7 samples were consistently negative. Blood samples from negative control sheep (sham-infected, or exposed to BSE, but uninfected) consistently produced negative results in repeat PMCA experiments. Thus, the majority of positive PMCA results were found in samples from sheep whose blood contained sufficient infectivity to transmit infection by transfusion.

To determine at what stage of infection blood samples from BSE-infected sheep would test positive by PMCA, longitudinal series of buffy coat samples collected from individual donor and recipient sheep were analysed. In donor sheep, PrP^Sc was detected in blood by PMCA from 4 months post-infection, although the timing of the onset of detection of PrP^Sc in individual sheep was very variable, (Fig 3A and Table 6). In contrast, in recipient sheep all but one sample collected at 6 months post-infection (the earliest sampling point for recipients), and all samples from 9 months post-infection onwards, gave positive PMCA results (Fig 3B and Table 7). Following the onset of detection, all subsequent samples collected from the same individual sheep tested positive in PMCA. These results indicate that the route and method of

**Table 6. Time course of detection of PrP^Sc by PMCA in blood samples from BSE-infected donor sheep.**

| Donor *PRNP* codon 141 genotype | PMCA results (number of positive results/number of samples tested) | | | | | | | |
|---|---|---|---|---|---|---|---|---|
| | Months post-infection (mpi) | | | | | | | |
| | 2 | 4 | 6 | 8 | 10 | 12 | 18 | Terminal* |
| LL | 0/4 | 2/4 | 3/4 | 3/4 | 4/4 | 4/4 | 2/2 | 4/4 (18–21 mpi) |
| FF | 0/3 | 1/3 | 1/3 | 1/3 | 1/3 | 2/3 | 3/3 | 3/3 (27–28 mpi) |
| LF | 0/3 | 1/3 | 1/3 | 1/3 | 1/3 | 3/3 | 3/3 | 3/3 (31–39 mpi) |
| **Total:** | 0/10 | 4/10 | 5/10 | 5/10 | 6/10 | 9/10 | 10/10 | 10/10 |

* The terminal time point is when sheep were culled following the onset of clinical disease, or for welfare reasons–the range of survival periods for individual sheep are indicated in months post-infection (mpi).

**Table 7. Time course of detection of PrP^Sc by PMCA in blood samples from BSE-infected recipient sheep.**

| Component transfused | Recipient *PRNP* codon 141 genotypes | PMCA results (number of positive results/number of samples tested) | | | | | |
|---|---|---|---|---|---|---|---|
| | | Months post-infection (mpi) | | | | | |
| | | 6 | 9 | 12 | 15 | 18 | Terminal* |
| Whole blood | 1 X LL, 1 x FF, 2 x LF | 3/4 | 4/4 | 4/4 | 3/3 | 3/3 | 4/4 (16–40 mpi) |
| Buffy coat | 1 x LL, 3 x FF | 4/4 | 4/4 | 4/4 | 4/4 | 4/4 | 4/4 (20–30 mpi) |
| Plasma | 1 x FF, 2 x LF | 3/3 | 3/3 | 3/3 | 3/3 | 3/3 | 3/3 (20–36 mpi) |
| Red cells | 1 x FF, 2 x LF | 3/3 | 1/1 | 1/1 | 1/1 | 1/1 | 1/1 (33 mpi) |
| **Total:** | | 9/14 | 12/12 | 12/12 | 12/12 | 12/12 | 12/12 |

* The terminal time point is when sheep were culled following the onset of clinical disease, or for welfare reasons–the range of survival periods for individual sheep are indicated in months post-infection (mpi).

infection can have a profound impact on the haematogenous dissemination of prions, and the sensitivity of detection of PrP^Sc in preclinical blood samples.

Due to the variation in incubation period and transfusion transmission efficiency associated with *PRNP* genotype (see above), we also examined whether *PRNP* codon 141 genotype influenced the outcome of PMCA analysis on the longitudinal blood sample series. Although there is no equivalent polymorphism to L141F in the human *PRNP* gene, the *PRNP* M129V polymorphism is associated with variation in the incubation period of acquired human prion diseases, and therefore may have similar effects on transmission of vCJD by transfusion. In donor sheep, one animal of each *PRNP* genotype (141LL, 141FF, 141LF) tested positive at 4 months post-infection. However, by 10 months post-infection (the time point at which blood was collected for transfusion), all four 141LL sheep tested positive by PMCA, compared to only one of three 141FF and 141LF sheep, respectively (Table 6). This result corresponds with the transfusion data, which showed significantly higher transmission rates for 141LL donors (Table 4 and Fig 2). In recipient sheep, the single sample that tested negative at 6 months post-infection was from a 141LF sheep that had a particularly long incubation period (40 months); otherwise *PRNP* codon 141 genotype had no effect on PMCA results (Table 7).

## Discussion

This study represents the largest experimental analysis to date of the prion infection risks associated with transfusion of labile blood components commonly used for treatment in human patients. The results confirm that blood transfusion can be a highly efficient route of transmission for prion diseases, and show that RBCs, platelets and fresh plasma from infected individuals are all potentially infectious, even following leucodepletion.

Data from sheep transfusion experiments, and bioassay of blood components in transgenic mice, indicated that the highest levels of infectivity were associated with the buffy coat fraction, and therefore are likely to be associated with WBCs. Our aim was to produce blood components that met the specifications of human transfusion services, rather than purified cell populations or cell-free plasma, so that each component contained varying proportions of plasma, platelets, WBCs and/or RBCs. However, a substantial proportion of infectivity in RBCs, platelets and plasma is probably associated with WBCs, since leucodepletion (reducing WBCs to $<10^6$ cells/unit transfused) significantly reduced the transmission rates of these components. The effect of leucodepletion appeared greatest for platelets, especially when considering the recipients of paired non-leucodepleted and leucodepleted components from the same donor (Table 3). This may be because platelet concentrates were prepared by centrifugation of platelet-rich plasma, therefore most of the WBCs present would have co-sedimented with platelets.

However, due to the small group sizes, there was insufficient power to carry out statistical tests comparing individual leucodepleted and non-leucodepleted components, and thus these results should be interpreted with caution. Since transmission of scrapie in sheep has been demonstrated following transfusion of $10^5$ WBCs [33], infectivity of leucodepleted components may still be attributable to WBCs, but we cannot exclude contributions from platelets, cell membrane fragments and plasma-associated soluble infectivity.

These results add to cumulative data from rodent and ruminant animal models indicating that the highest levels of prion infectivity in blood are associated with WBCs [7,9,12,32,34]. Infectivity has also been demonstrated in purified platelets from sheep and deer experimentally infected with scrapie and CWD [34,35], respectively, whereas platelets from scrapie-infected hamsters contained negligible infectivity [36]. Whilst plasma has been estimated to contain 40%-60% of blood infectivity in rodent prion models [8,9,32], the data presented in this paper, and from studies in scrapie-infected sheep and CWD-infected deer, suggest much lower levels of infectivity in plasma [12,34,37].

Leucodepletion of blood components is an important component of precautionary measures introduced by blood transfusion services to reduce or prevent transmission of vCJD by blood products. The Transfusion Medicine Epidemiology Review (TMER) has so far identified three cases of vCJD in recipients of non-leucodepleted RBCs from donors who later developed vCJD, but no cases in recipients of leucodepleted RBCs [13]. Whilst leucodepletion significantly reduced transfusion transmission rates in our model, we found that red cells, platelets and plasma could all still transmit infection following leucodepletion. Similarly, a study in scrapie-infected sheep showed limited transmission of infection following transfusion of leucodepleted RBCs and plasma [38]. Interestingly, a small number of macaques transfused with leucodepleted RBCs from vCJD-infected donors developed an unusual myelopathic neurological disorder, suggesting that transmission had occurred but resulted in an altered clinical phenotype [39]. We did not identify a similar altered clinical phenotype in our transfusion recipients. Unlike the macaques, the idiopathic ataxia observed in some sheep affected both orally infected donors and transfusion recipients. Collectively, these results strongly support the continued use of leucodepletion to prevent vCJD transmission by blood transfusion, but suggest that risks could be further reduced by improving methods/devices for removal of infectious material from blood.

The period during which prion infectivity can be detected in the blood of an infected individual, prior to developing signs/symptoms of disease, is an important factor in determining the risks of transmission by blood transfusion. In this study, blood was collected from donor sheep at a preclinical time point (10 months post-infection) approximately one-third to half-way through the survival period of individual animals. Less than half of BSE-infected donors (15/33) transmitted infection to one or more transfusion recipients at this point, but our earlier experiments and other studies suggest that transmission rates are likely to progressively increase when donors reach the later preclinical stages of infection [11,35]. The probability of transfusion transmission at 10 months post-infection was associated with the donor *PRNP* codon 141 genotype, with the highest transmission rates for donors homozygous for leucine (141LL) at this position. This is most likely due to the variation in survival period associated with the *PRNP* L141F polymorphism [30], such that donors with longer survival periods (141FF and 141LF) were at earlier stages of incubation relative to disease onset at the time of donation, compared to 141LL donors. Although there is no equivalent polymorphism to L141F in the human *PRNP* gene, the *PRNP* M129V polymorphism is associated with variation in the incubation period of acquired human prion diseases [40], and therefore may have a similar effect on the period during which individuals are potentially capable of transmitting infection to others.

We also demonstrated that sheep infected with BSE by blood transfusion are equally, if not more, likely than orally infected donors to transmit infection by transfusion. When one unit of whole blood was collected from primary recipients at 15 months post-infection and transfused into secondary recipients, the transmission rate was 100% for infected donors. This indicates that acquisition of prion infection by blood transfusion does not lead to attenuation of the infection. If this reflects the situation in human vCJD, then transfusion could contribute to ongoing human-to-human transmission, depending on the prevalence of infection in blood donors and susceptibility to disease in recipients of blood products.

To examine the relationship between the stage of infection and the presence of infectivity in blood in more detail, we employed a sensitive PMCA assay to test for the presence of PrP$^{Sc}$ in longitudinal series of blood samples collected at intervals from donor and recipient sheep. Initially, it was important to establish whether PMCA results reflected the presence of infectivity, as there is not always a direct correlation between PrP$^{Sc}$ concentration and titres of infectivity in tissues/body fluids. By testing blood samples shown to be infectious or non-infectious in the sheep transfusion experiments, we found that the majority of infectious samples tested positive by PMCA, while the majority of non-infectious samples were PMCA negative. In addition, following serial titration of blood components used to seed PMCA, the highest levels of PrP$^{Sc}$ were found in buffy coat samples and the lowest in plasma (S2 Fig), which reflects the infectivity data from sheep transfusion and mouse bioassay experiments.

PMCA was able to detect PrP$^{Sc}$ in preclinical blood samples as early as 4 months post-infection in orally infected donor sheep, but there was marked variability in the time point at which blood from individual donors first tested positive (range 4–18 months post infection). In contrast, samples from all but one of fourteen recipients infected by transfusion tested positive by PMCA at 6 months post-infection (the earliest sample collection time for this group). It is therefore likely that a significant proportion of recipients would also have given positive PMCA results at earlier time points. This result further emphasizes the efficiency of blood transfusion in transmitting prion infection, and demonstrates that the route and method of infection can have a profound effect on dissemination of prions in the bloodstream. This in turn influences the likelihood of transmitting infection; although sheep infected with BSE orally or by transfusion may have equally high transmission rates if blood was donated at 15 months post-infection, the PMCA data suggest that sheep infected by transfusion would have transmitted infection much more efficiently than orally infected sheep at earlier time points.

PMCA shows promise as a potential blood-based diagnostic test for vCJD, with different versions of the assay demonstrating 100% specificity and 100% sensitivity in identifying samples from vCJD patients [41,42]. However, the ability to detect individuals during the preclinical, asymptomatic phase of infection is critical to the utility of any potential diagnostic test, and can only be assessed using animal models, due to the lack of sufficient preclinical samples from vCJD cases. Our results suggest that it is important to consider the route of infection when evaluating the sensitivity of diagnostic tests on preclinical samples. During the early preclinical phase of infection (6–12 months post-infection), the overall sensitivity of detection of samples from BSE-infected sheep by PMCA was much lower in orally (63%) compared to intravenously (98%) infected sheep. Previous studies have demonstrated detection of PrP$^{Sc}$ in blood by PMCA from 6 months post-infection in sheep orally infected with BSE [29] and as early as 2 months post-infection (~25 months before clinical onset) in cynomolgus macaques experimentally infected with vCJD by intravenous/intraperitoneal routes [43]. Although collectively these results show that PMCA assays are capable of detecting prion-infected individuals at early preclinical time points, our data suggest that further improvements in assay sensitivity might be necessary for reliable detection of preclinical/subclinical vCJD in the human population, where oral exposure to variable doses of BSE is the most likely route of infection.

In view of the highly efficient transmission of prion diseases by blood transfusion demonstrated in sheep, including transmission by leucodepleted components, it is perhaps surprising that there have not been more transfusion-associated cases of vCJD. The TMER study in the UK has not identified any new cases of vCJD (or evidence of infection) since 2006 in continuing follow-up of recipients of blood components from donors who later developed vCJD, with 13 individuals known to have survived for >10 years after transfusion [13]. Our experimental studies used genetically susceptible animals, with controlled doses and timing of infection, and known survival periods. In contrast, susceptibility to infection, source of exposure, infectious dose, stage of infection in the donor, and relative titres of infectivity in blood could all vary widely in the human population, and this might explain the apparent discrepancy in outcomes. Nevertheless, the results of this study indicate that the potential risk of transmission of vCJD by transfusion may be higher than previously thought, even when blood components are leucodepleted.

In summary, our results highlight the importance of continuing current measures to prevent transfusion transmission of vCJD (e.g. leucodepletion, donor deferral), but suggest that there may be room for improvement. Risks could potentially be further reduced by enhanced methods of prion removal from blood products, and development of ultrasensitive assays for preclinical detection of prions, to be deployed in screening blood donors.

## Supporting information

**S1 Table. Donor sheep—clinical status, survival times and results of IHC and Western blotting on tissues.**
(PDF)

**S2 Table. Primary recipient sheep—clinical status, survival times and results of IHC and Western blotting on tissues.**
(PDF)

**S3 Table. Titration of BSE sheep brain inoculum (BSB/5/03) in TgshpXI mice.** The inoculum was prepared from a single sheep infected with BSE by intracerebral inoculation of BSE-infected cattle brain homogenate. Each mouse was inoculated intracerebrally under general anaesthesia with 25 μl of the brain dilution indicated, and monitored for development of clinical signs up to 700 days post infection. Brains from all mice were tested for the presence of PrP$^{Sc}$ using an ELISA kit (BetaPrion BSE EIA Test Kit; AJ Roboscreen Gmbh, Leipzig, Germany) to confirm infection status.
(DOCX)

**S4 Table. Details of outcomes of secondary blood transfusions.**
(DOCX)

**S5 Table. Secondary recipient sheep—clinical status, survival times and results of IHC and Western blotting on tissues.**
(PDF)

**S6 Table. Infectivity detected by bioassay in TgshpXI mice in sheep blood during the time course of BSE infection.** Each mouse was inoculated intracerebrally under general anaesthesia with 25μl of the indicated components, diluted 1:1 in sterile PBS (dilution was necessary due to acute toxicity associated with inoculation of blood components collected with EDTA as anticoagulant), and monitored for development of clinical signs up to 700 days post infection. Brains from all mice were tested for the presence of PrP$^{Sc}$ using an ELISA kit (BetaPrion BSE EIA Test Kit; AJ Roboscreen Gmbh, Leipzig, Germany) to confirm infection status. There

were no transmissions in groups of mice injected with samples from time points 0, 4 and 8 months post-infection (data not shown). Infectious titres (ID/ml) were calculated by limiting dilution titration (where distribution of infectivity into individual inoculations is assumed to follow a Poisson distribution). NA–not applicable.
(DOCX)

**S7 Table. Detection of PrP$^{Sc}$ by PMCA in buffy coat and plasma samples from donor sheep in the clinical phase of BSE infection.** Samples of buffy coat and plasma (undiluted and diluted 1:1 with PMCA buffer) from 4 donor and 10 recipient sheep clinically affected with BSE and 4 negative controls were used to seed PMCA reactions (without added dextran sulphate). The table shows the numbers of positive PMCA reactions after up to 6 rounds of serial PMCA. All buffy coat samples gave positive results, while only 4/14 reactions seeded with plasma were positive.
(DOCX)

**S1 Fig. Optimization of PMCA for detection of PrP$^{Sc}$ in BSE-infected sheep blood.** A. To assess the potential inhibitory effects of blood components, PMCA reactions were seeded with tenfold serial dilutions ($10^{-4}$ to $10^{-11}$) of BSE-infected sheep brain homogenate diluted in either PMCA buffer or uninfected buffy coat. The limit of detection (brain dilution of $10^{-8}$) was the same for both dilution series, and was reached after 3 rounds (R3) of serial PMCA. C–positive control; BSE-infected sheep brain homogenate (no PK digestion; 0.36mg brain equivalent). B. Dextran sulphate was added to give a final concentration of 0.5% (w/v) in PMCA reactions seeded with tenfold serial dilutions of the same BSE-infected sheep brain homogenate. The limit of detection (brain dilution of $10^{-9}$) was reached after 2 rounds (R2) of PMCA in the presence of dextran. C–positive control; BSE-infected sheep brain homogenate (PK digested; 1.7mg brain equivalent).
(TIF)

**S2 Fig. Detection of PrP$^{Sc}$ in blood components from BSE-infected sheep by PMCA.** Samples of whole blood, buffy coat and plasma from two BSE-infected donor sheep (M490 –upper panel; N236 –lower panel) at the clinical stage were used to seed PMCA reactions either undiluted (UN) or following dilution in PMCA buffer as indicated (1:1, 1:4, tenfold serial dilutions). The image shows the results after 6 rounds (R6) of serial PMCA (no added dextran), demonstrating positive amplification in all blood fractions with varying sensitivity. C–positive control; scrapie-infected Rov9 cell lysate (PK digested; 37.5 μg total protein equivalent). B–blank lane. NC1, NC2 –negative control samples (buffy coat and mesenteric lymph node, respectively) from mock-infected sheep.
(TIF)

## Acknowledgments

The authors thank past and present members of large animal research services at the Institute for Animal Health, Compton and the Roslin Institute for excellent care of the sheep and technical assistance with experimental procedures. We acknowledge the contribution of Hugh Simmons and colleagues from the Animal and Plant Health Agency (APHA) and ADAS in breeding and provision of sheep for the experiment. We would also like to thank Wilfred Goldmann for advice and support with *PRNP* genotype analysis, Dawn Drummond and colleagues at Easter Bush pathology for processing tissues for histopathology, and colleagues at SNBTS for preparation and provision of blood bags. The BSE-infected and normal cattle brain homogenates were provided by the Biological Archive Group (formerly TSE Archive) at APHA Weybridge.

## Author Contributions

**Conceptualization:** Martin H. Groschup, Marc Turner, E. Fiona Houston.

**Data curation:** M. Khalid F. Salamat, A. Richard Alejo Blanco, Sandra McCutcheon, Kyle B. C. Tan, Paula Stewart, Allister Smith, Christopher de Wolf, Dietmar Becher, E. Fiona Houston.

**Formal analysis:** M. Khalid F. Salamat, A. Richard Alejo Blanco, Sandra McCutcheon, Helen Brown, E. Fiona Houston.

**Funding acquisition:** Sandra McCutcheon, Marc Turner, Jean C. Manson, E. Fiona Houston.

**Investigation:** M. Khalid F. Salamat, A. Richard Alejo Blanco, Sandra McCutcheon, Kyle B. C. Tan, Paula Stewart, Allister Smith, Christopher de Wolf, Dietmar Becher, E. Fiona Houston.

**Methodology:** Sandra McCutcheon, Martin H. Groschup, Olivier Andréoletti, E. Fiona Houston.

**Project administration:** Sandra McCutcheon, Jean C. Manson, E. Fiona Houston.

**Resources:** Martin H. Groschup, Olivier Andréoletti, Marc Turner.

**Software:** Christopher de Wolf.

**Supervision:** A. Richard Alejo Blanco, Sandra McCutcheon, Martin H. Groschup, Jean C. Manson, E. Fiona Houston.

**Writing – original draft:** M. Khalid F. Salamat, A. Richard Alejo Blanco, Sandra McCutcheon, Helen Brown, E. Fiona Houston.

**Writing – review & editing:** M. Khalid F. Salamat, A. Richard Alejo Blanco, Sandra McCutcheon, Kyle B. C. Tan, Paula Stewart, Martin H. Groschup, Marc Turner, Jean C. Manson, E. Fiona Houston.

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
