## [Decision Letter · Decision Letter 0]

28 Nov 2020

Dear Dr Houston,

Thank you very much for submitting your manuscript "Preclinical transmission of prions by blood transfusion is influenced by donor genotype and route of infection." for consideration at PLOS Pathogens. As with all papers reviewed by the journal, your manuscript was reviewed by members of the editorial board and by several independent reviewers. The reviewers appreciated the attention to an important topic. Based on the reviews, we are likely to accept this manuscript for publication, providing that you modify the manuscript according to the review recommendations.

Please be mindful of comments from reviewer's 1 and 2 regarding clarity, additionally, pay special attention to point 3 by reviewer 3. Thank you.

Sincerely,

Jason C Bartz

Associate Editor

PLOS Pathogens

Michael Malim

Section Editor

PLOS Pathogens

Kasturi Haldar

Editor-in-Chief

PLOS Pathogens

orcid.org/0000-0001-5065-158X

Michael Malim

Editor-in-Chief

PLOS Pathogens

orcid.org/0000-0002-7699-2064

Please be mindful of comments from reviewer's 1 and 2 regarding clarity, additionally pay special attention to point 3 by reviewer 3. Thank you.

Reviewer Comments (if any, and for reference):

Reviewer's Responses to Questions

**Part I - Summary**

Reviewer #1: It is well documented that blood components are capable of transmitting prion disease during the preclinical phase of infection, and that concerns about the safety of human blood products remain. These concerns arose from retrospective surveys revealing that an estimated 1 in 4000 to 1 in 2000 individual in the UK may be subclinically infected (i.e. they will never develop symptoms of vCJD within their natural lifespan), and that their potential to transmit infection by blood transfusion or other iatrogenic routes is currently unknown. These findings are compounded by demonstration of vCJD infections in individuals with a PRNP genotype that has been associated with prolonged incubation periods in other acquired human prion diseases (e.g. kuru, iatrogenic CJD), heightening concerns of a possible second wave of vCJD cases.

Studies to unravel the dynamics of transmission during the asymptomatic phase of disease, as well as the implications of oral or blood-borne infections and PRNP genotype on infection rates must be done in animal systems the recapitulate human prion disease. Scrapie infections in sheep and chronic wasting disease (CWD) infections in cervids demonstrate remarkable similarities to VCJD in humans.

The manuscript is very well organized and written. The authors have provided ample evidence to support their conclusions.

This study represents the largest and longest experimental analysis to date of the risks of prion infection associated with transfusion of labile blood components, and the effectiveness of leucodepletion in preventing transmission. Using a BSE-infected sheep model, they found that red blood cells, platelets and plasma from preclinical donors were all infectious, even after leucodepletion, although leucodepletion significantly reduced transmission rates. In addition, the time course of detection of prions in blood varied significantly depending on the route and method of infection.

They show that infectivity titers in blood components reflect transfusion transmission efficiency, and report on the outcomes of secondary transfusions (with primary recipients as donors). In addition, they demonstrate amplification competent PrPSc detection in blood by protein misfolding cyclic amplification assay (PMCA) over the longitudinal course of infection.

Reviewer #2: This study summarizes a long-term experiment designed to determine the risk of prion transmission upon transfusion of blood and blood components that are relevant in human medicine. A model of BSE in sheep was used, and upon initial oral infection of sheep with BSE infected brain homogenate, at 10 months post infection, blood and blood components as well as leucodepleted blood components were used for transfusion in recipient sheep. These experiments confirmed previous findings that most prion infectivity is associated with white blood cells, and point out that leucodepletion reduces, but does not eliminate, prion infectivity of blood. However, efficiency of these primary transmission was dependent on the donor Prnp genotype at codon 141, which affects the incubation period. In a secondary transmission experiment, whole blood from the primary recipients was transfused to secondary recipients, resulting in a 100% attack rate. Altogether, these studies provide novel insights into the efficiency of blood transfusion and highlight the influence of the donor’s Prnp genotype and the timing of blood collection for transfusion on the efficiency of prion transmission.

Reviewer #3: In the manuscript entitled “Preclinical transmission of prions by blood transfusion is influenced by donor genotype and route of infection”, the authors described a wide and consequent long-term study on the transmission of BSE through transfusion of blood and blood components in the experimental model of BSE-infected sheep. This study constitutes the largest experimental transfusion study in a pertinent model of human situation, and these results are of high importance in the prion field and have important consequences for diagnostic and prevention purposes towards the transfusional risk of vCJD.

**Part II – Major Issues: Key Experiments Required for Acceptance**

Reviewer #1: N/A

Reviewer #2: The blood samples for transfusion were taken at 10 months post infection from donors and 15 months post infection from primary recipients. PMCA results show that blood samples of only 6/10 donors (orally infected with BSE) were positive, whereas after 12 months 9/10 and after 18 months 10/10 were positive. In secondary transfusion, blood samples for transfusion was obtained at 15 months, a time point when all samples that were analysed were positive in PMCA. Therefore, the authors should consider adding to the discussion whether the sheep infected by transfusion are indeed more likely to transmit infection, as presumably at 15 months post infection, transfusion of blood from orally infected sheep might have resulted in 100% transmission.

Reviewer #3: Despite the quality of the work and the soundness of the results, I have three main concerns

1) The authors mentioned lines 265-267 that “The majority of donor sheep (33/39) orally dosed with BSE-infected brain homogenate developed typical clinical signs, and were confirmed as being infected by detection of PrPsc in tissues tested post mortem by Western blotting and/or immunohistochemistry (IHC)”. They announced that brain, tonsil, spleen, Peyer’s patches, mesenteric lymph nodes and prescapular nodes were culled. Did the authors test all the samples for all the animals, including for the negative ones?

The authors also defined “evidence of BSE infection” as “clinical signs and/or positive results from western blotting and IHC” (lines 277-278). The criteria of inclusion are clearly unclear, since they may literally include several situations, ranging from a sheep under incubation with only one positive follicle to clinically-affected animals at the terminal stage of the disease. Moreover, incubation period is defined as the interval between the date of infection and the date on which the animal was culled, and thus include the clinical period that is very different upon animals and dependent on the moment the authors decided to proceed to the euthanasia. What is the duration of the clinical period for each animal? Based on the pattern of information provided in this manuscript, it is impossible to draw comparisons between animals according to their status (Infection? Disease?) and their incubation period, which is a capital information in prion diseases since it is reputed to be proportional to the initial infectious dose. For each animal presented in this study, a detailed table should be provided with individual information about clinical signs and biochemical / IHC positivity in all the tested organs. Moreover, information should be provided upon the cause of death for negative animals, including the three animals mentioned as ataxic (line 268). The authors mentioned the occurrence of altered clinical phenotypes in a macaque model of transfusion (line 501); did they observe similar observations in this sheep model?

2) As mentioned above, incubation period is expected to be proportional to the initial infectious dose. In this study the relation between these two parameters is apparently incoherent since bioassay in transgenic mice allowed the estimation of infectivity titers, which is completely unrelated to the incubation period in sheep: for example, buffy coat of N236 is estimated 13.1 ID/ml and transmit disease in 904 days, whereas buffy coat of N257 is estimated 2.8 ID/ml and transmit disease in 639 days. Is there a difference in terms of injected volume between animals? The authors should discuss this issue. The authors mentioned that all the samples were tested by PMCA; a reliable comparison between bioassay in mice, PMCA and transmission to sheep would have been expected. Moreover, BSE was transmitted to several sheep after plasma transfusion: the authors should discuss why no bioassay in mice was positive with plasma samples.

3) One main conclusion of the paper relies on the efficiency of leucodepletion: the authors underline the significant differences between the rates of BSE transmission after transfusion of blood products (plasma, platelet and RBCC) and their leukodepleted counterparts. However, one could have another lecture of these results. Indeed, when transmission did not occur with non-leucodepleted blood products (plasma for N245, N189, N204, N231, N157, N180; platelets for N231 and N157; RCC for N189, N204 and N231), the absence of transmission after leucodepletion is also expected. Thus, one may focus on the effect of leucodepletion only for blood products that transmitted infection, and the efficiency of leucodepletion would be analyzed for paired subjects.

In this context, the analysis of leucodepletion on plasma is limited to donors N233, N261 and N232. Leucodepletion was efficient only on N232 plasma (no transmission after leucodepletion), whereas leucodepletion effect was weak (N261, doubling of incubation period) to null (N233, same incubation periods) for the two other plasmas. For RCC (limited to N233, N261, N232, N157 and N181), leucodepletion was efficient for only two samples (N157 and N180 donors), but inefficient for the three other ones (N233, N261 and N232). For PLT, whereas N261 should not be considered (recipient of leukodepleted PLT died after 33 days), leucodepletion seemed efficient since five recipients among 6 did not develop disease.

According to their results, the authors should modulate their conclusions on the efficiency of leucodepletion, that is finally mostly focused on platelets, and they should discuss these differences of efficiency; do the authors think that these differences might be linked to the plasmatic soluble infectivity that would be differently distributed among the types of blood products?

**Part III – Minor Issues: Editorial and Data Presentation Modifications**

Reviewer #1: 1) No IHC data is shown, negating the need for a methods section.

2) Clearly denote the number of rounds required for PMCA detection throughout the manuscript, including abbreviations, i.e. (R) used in some cases, Round in others.

3) Lines 279 and 280: Each line mentions ‘this figure’, yet I believe the authors are referring to Table 1. Please clarify.

4) Lines 298-303: Refers to a BSE brain mouse bioassay titration. Please provide data or publication reference, as this titration was used to estimate infectious doses for blood components.

5) Tables 1 and 3. For consistency please include % Neg.

6) Line 354: “At the time of blood collection, it was not possible to determine which of the primary recipients acting as donors in this experiment were infected with BSE”. Verifying that a RAMALT biopsy was not collected to provide scrapie PrPSc deposition status.

7) Line 391: “The most likely explanation for these results is that the probability of transmission is related to the stage of the incubation period reached by donor sheep at the time of blood collection”

a. Please clarify what is meant by “stage of incubation period”. Do the authors mean infectivity titers in blood may be too low to initiate infection, or that prions may be absent from blood all together?

b. PMCA blood analysis did not include dilutional series, thus cannot provide information about titers in blood. Thus could lines 391 and 398 suggest again titer vs stage of disease?

8) Fig. Legends and Fig. Titles: Seeding activity vs PrPsc. Please address for consistency.

Reviewer #2: (No Response)

Reviewer #3: (No Response)

PLOS authors have the option to publish the peer review history of their article (what does this mean?). If published, this will include your full peer review and any attached files.

Reviewer #1: No

Reviewer #2: No

Reviewer #3: No
---

## [Editor Report · Decision Letter 1]

4 Jan 2021

Dear Dr Houston,

We are pleased to inform you that your manuscript 'Preclinical transmission of prions by blood transfusion is influenced by donor genotype and route of infection.' has been provisionally accepted for publication in PLOS Pathogens.

Best regards,

Jason C Bartz

Associate Editor

PLOS Pathogens

Michael Malim

Section Editor

PLOS Pathogens

Kasturi Haldar

Editor-in-Chief

PLOS Pathogens

orcid.org/0000-0001-5065-158X

Michael Malim

Editor-in-Chief

PLOS Pathogens

orcid.org/0000-0002-7699-2064

The authors have satisfactorily addressed the concerns of the reviewers
---

## [Editor Report · Acceptance letter]

25 Jan 2021

Dear Dr Houston,

We are delighted to inform you that your manuscript, "Preclinical transmission of prions by blood transfusion is influenced by donor genotype and route of infection.," has been formally accepted for publication in PLOS Pathogens.

Best regards,

Kasturi Haldar

Editor-in-Chief

PLOS Pathogens

orcid.org/0000-0001-5065-158X

Michael Malim

Editor-in-Chief

PLOS Pathogens

orcid.org/0000-0002-7699-2064